# Cortisol Interaction with Aquaporin-2 Modulates Its Water Permeability: Perspectives for Non-Genomic Effects of Corticosteroids

**DOI:** 10.3390/ijms24021499

**Published:** 2023-01-12

**Authors:** Robin Mom, Stéphane Réty, Daniel Auguin

**Affiliations:** 1Research Group on Vestibular Pathophysiology, CNRS, Unit GDR2074, F-13331 Marseille, France; 2Laboratoire de Biologie et Modélisation de la Cellule, ENS de Lyon, University Claude Bernard, CNRS UMR 5239, INSERM U1210, 46 Allée d’Italie Site Jacques Monod, F-69007 Lyon, France; 3Laboratoire de Biologie des Ligneux et des Grandes Cultures, Université d’Orléans, UPRES EA 1207, INRA-USC1328, F-45067 Orléans, France

**Keywords:** AQP2, cortisol, corticosteroid, non-genomic effects, molecular dynamics, water permeability

## Abstract

Aquaporins (AQPs) are water channels widely distributed in living organisms and involved in many pathophysiologies as well as in cell volume regulations (CVR). In the present study, based on the structural homology existing between mineralocorticoid receptors (MRs), glucocorticoid receptors (GRs), cholesterol consensus motif (CCM) and the extra-cellular vestibules of AQPs, we investigated the binding of corticosteroids on the AQP family through in silico molecular dynamics simulations of AQP2 interactions with cortisol. We propose, for the first time, a putative AQPs corticosteroid binding site (ACBS) and discussed its conservation through structural alignment. Corticosteroids can mediate non-genomic effects; nonetheless, the transduction pathways involved are still misunderstood. Moreover, a growing body of evidence is pointing toward the existence of a novel membrane receptor mediating part of these rapid corticosteroids’ effects. Our results suggest that the naturally produced glucocorticoid cortisol inhibits channel water permeability. Based on these results, we propose a detailed description of a putative underlying molecular mechanism. In this process, we also bring new insights on the regulatory function of AQPs extra-cellular loops and on the role of ions in tuning the water permeability. Altogether, this work brings new insights into the non-genomic effects of corticosteroids through the proposition of AQPs as the membrane receptor of this family of regulatory molecules. This original result is the starting point for future investigations to define more in-depth and in vivo the validity of this functional model.

## 1. Introduction

Aquaporins (AQPs) are transmembrane water channels [1,2,3] found in almost all living cells [4] and are the major effectors of the regulation of cell homeostasis and trans-cellular water fluxes. AQPs are subject to a myriad of fine regulations, such as post-transcriptional modifications [3,5], phosphorylation-dependent vesicular trafficking [6,7], associations with other proteins [8,9] and by gating mechanisms [10,11,12]. These proteins are implicated in several pathophysiologies and were, hence, considered as putative drug targets [3,13,14,15]. Multiple attempts have been made or are still underway to develop drugs targeting AQPs without very significant advances so far [16,17]. Our previous work, through in silico approaches, suggests that the corticosteroid dexamethasone could specifically bind to the extra-cellular surface of human AQP2 and impair the channel water permeability significantly. This modulation of water fluxes was accomplished through the modulation of the size and electrostatic profile of the narrowest constriction of the AQP channel, which is the aromatic/arginine (ar/R) constriction. This interaction impacted the water permeability the most significantly when dexamethasone was directly bound to the very conserved arginine of the constriction (R187 in AQP2) [18]. However, the effects of other corticosteroids on AQPs have not been explored.

Corticosteroids constitute a wide family of natural and synthetic hormones with a broad spectrum of actions on human physiology [19,20]. Because of the hydrophobic nature of corticosteroids, their classical mode of action involves their passive diffusion through plasma membrane and binding in the cytoplasm to dedicated nuclear receptors (glucocorticoid and mineralocorticoid receptors GR and MR, respectively [21,22,23]), which then translocate in the nucleus to regulate gene expression. Indeed, these nuclear receptors are transcription factors whose regulatory function is activated by the binding of corticosteroids [24,25,26]. The induced genomic regulation triggers physiological responses visible at the soonest after a 15 min time delay [27]. However, it appeared that some corticosteroid effects could be induced in a very short time (less than 3 min) and were not affected by GR or MR blocking or protein synthesis inhibition (for a review see [27]). Therefore, a fast non-genomic pathway exists in parallel to long-term genomic gene regulation, suggesting the existence of membrane receptors mediating at least part of these non-genomic effects (also known as rapid steroid effects) (reviewed in [28,29]). In many cases, the non-genomic corticosteroid response is linked to the regulation of cell homeostasis and of cell volume; therefore, we questioned if AQPs could be involved in non-genomic corticosteroid response. In the present study, we investigated the AQP family as a putative corticosteroid membrane receptor. In order to do so, based on similarities shared between GR, MR, cholesterol consensus motif (CCM) and the extracellular-surface of AQPs, we proposed a first putative AQPs corticosteroid binding site (ACBS). We delimited this ACBS as spanning from the extra-cellular arginine of the ar/R constriction up to the end of the extra-cellular vestibules of AQPs, where flexible loops (such as C-loop and A-loop) are found. We then evaluated the impact of the interaction of the naturally produced stress hormone cortisol with human AQP2 and detailed the molecular consequences on the water channel properties. Finally, we discussed the physiological relevance of such an AQPs corticosteroid binding site (ACBS) in the light of the non-genomic effects of corticosteroids.

## 2. Results

### 2.1. Prediction of the AQPs Corticosteroid Binding Site (ACBS) through Amino Acid Sequences and Structural Alignment

In a previous study, we hypothesized through in silico molecular dynamics, that AQP2 could interact with the glucocorticoid dexamethasone and that this interaction would have a significant inhibitory effect on the water permeability of the channel [18]. From these first results and based on the similarities between the AQPs extracellular vestibules with MR, GR and the CCM, we propose here a putative AQPs corticosteroid binding site (ACBS).

To define an AQPs corticosteroid binding site (ACBS), portions of trajectory from AQP2—dexamethasone and AQP2—cortisol simulations (experimental setup 3, see methods) corresponding to stable hydrogen bond interaction between R187 (of the ar/R constriction) and the corticosteroid were used (Figure 1A). This criterion was chosen based on the observations made in our previous study, indicating this interaction is a good marker of water permeability inhibition and is of a spontaneous and recurrent nature [18]. First of all, it is comforting to see that the residues establishing hydrogen bonds with the corticosteroids are conserved between the AQP2—dexamethasone and the AQP2—cortisol simulations (Figure 1B). Moreover, the AQP2 vestibule shares some conserved properties of MR and GR interaction pockets: the ketone group of the corticosteroid is stabilized by an arginine (R817, R611 and R187 for MR, GR and AQP2, respectively) and a polar residue (Q776 and Q570 for MR and GR, respectively, and N119 for AQP2) [30,31]; the hydroxyl groups are in interaction with other polar residues: N770 and T945 for MR or N564 and T739 for GR [30]. Such polar residues are found at a similar location in the AQP2 corticosteroid interaction site (Q36, S122, N123, H177 or W34 Figure 1C). Cholesterol is a precursor in the synthesis of corticosteroids and, hence, shares with them the same sterol ring. Interestingly, the cholesterol consensus motif (CCM), found in 44% of human class A receptors [32], is also strictly conserved in human AQP2 (Figure 1C). This CCM is composed of four sites. Position 1 must be composed by an arginine or a lysine and corresponds to R187 in AQP2. This position accommodates the polar head of cholesterol manifested by the ketone group. Position 2 contributes to hydrogen bonds and is the most conserved position with a tryptophan in 94% of class A receptors and is found in AQP2 as well (W34) (Figure 1B,C) [32]. Position 3 must be made of an isoleucine, a valine or a leucine and is represented by I44 in AQP2 (Figure 1C). Position 4 can either be a phenylalanine or a tyrosine and corresponds to F48 in AQP2 (Figure 1C) [32].

In order to further investigate the relevance of such a putative AQPs corticosteroid binding site, we made a structural alignment of AQP2 with the experimental structures of four other human AQPs (Figure 2A,B). The AQP2 conformation used as the structural reference was retrieved from another simulated trajectory but with the same selection criterion, i.e., the formation of a hydrogen bond between ar/R arginine (R187) and the cortisol molecule for at least 50 consecutive nanoseconds (see methods). From this complementary set of analysis, we can observe that the residues corresponding to R187, N119 and I44 in AQP2 are functionally conserved between the AQPs (R187 is strictly conserved, I44 corresponds to residues with hydrophobic side chains and N119 to polar residues—except for AQP10 displaying a polar threonine at an adjacent position). These residues are both implicated into the formation of the CCM and an MR- or GR-like pocket around the ketone group of the corticosteroid. All are located on core structural features, which correspond to the most stable (Figure 2C) and the most conserved (Figure 2D) regions of the AQP fold. Other residues with hydrophobic side-chains and located on a similar part of the channel are also conserved (L28, A31, V41, A117 or V118) and further accommodate the lipidic nature of corticosteroids (Figure 2A,B). The residues with polar side-chains that could interact with the hydroxyl groups of the corticosteroid (Q36, W34, S122, N123 or H177) are, on the contrary, mainly located on variable and flexible loops, such as the as A-loop or C-loop (Figure 2C–E). Therefore, they are not strictly conserved on the structural alignment (Figure 2A–E). However, contrary to the MR and GR interaction pockets, the AQP extra-cellular vestibule displays a wide opening where free water molecules are present. These water molecules can accommodate the hydroxyl groups of the corticosteroid as well and compensate for the lack of a direct interacting residue. Finally, it appears that by mimicking both parts of the CCM and of MR and GR interaction pockets, the AQP extra-cellular vestibule could constitute a corticosteroid binding site. The core residues of this putative AQPs corticosteroid binding site are the most conserved structurally and between AQP homologs and are (for AQP2) R187, N119 and I44, fixating the corticosteroid through the hydrogen bond with its ketone and accommodating the lipidic nature of its sterol ring. This first half of the binding site can most likely interact with mineralocorticoids and glucocorticoids. On the other hand, polar residues involved in the stabilization of the hydroxyl groups of the molecule correspond to variable and flexible portions of the AQP and, hence, could be tied to specific differences between AQP/corticosteroid couples.

### 2.2. Cortisol Interaction with AQP2 Significantly Impairs Water Permeability

We then evaluated the impact of cortisol interaction with the AQP2 extra-cellular surface on the channel water permeability (Figure 3). In a first experimental setup, we mimicked the procedure published in our previous work [18] and manually docked the cortisol (COR) on the predicted ACBS inside the extra-cellular vestibules of AQP2. From the cumulative number of water molecules crossing the whole transmembrane section of the channel (Figure 3A), one can see the segregation between “control” and “COR” conditions. AQPs are naturally assembled in tetramers, with each of the four subunits (or chains) being a functional water channel. In the “control” condition, all chains display linear progressions, while in the “COR” condition, plateau phases are clearly visible, especially for chain D and chain B. When the four chains are pooled and the two conditions compared, traditional permeability indicator *pf* is impaired with a *p* value of 0.09 (Figure 3B). We already observed that *pf* could over-estimate the permeability of closed channels [12], hence, we also compared the two conditions with a more straightforward approach consisting of using the number of water molecules crossing the entire 30 angströms-long channel section as a permeability indicator. By using this alternative estimation of AQP2 water permeability, a very significant difference appears—*p* value of 2.3 × 10^−5^—between the two conditions (Figure 3B). However, when we look at the differences between “control” and “COR” for each chain (Figure 3C), we can see that some differences exist. The two chains with their channel water permeability impacted the most significantly are chain B and chain D. However, the corresponding water free energy profiles differ (Figure 3D). Chain D displays a free energy profile similar to what we observed for the AQP2–dexamethasone interaction: small free energy barriers are localized in all the conducting pore regions, indicating an effect of the interaction on the entire conducting pore [18]. On the other hand, the free energy profile of water inside chain B is characterized by a very high free energy barrier at the ar/R constriction in “COR”. This type of profile is similar to what can be observed in voltage-gating of AQPs when a conformational change in the arginine of the ar/R constriction induces an obstruction or a reduction in the pore diameter [12,33]. Finally, we observed an unexpected free energy profile for chain A “COR” with the disappearance of the free energy barriers in the extra-cellular half of the pore. However, this lowered free energy barrier in chain A did not correlate with a change in water permeability between the two conditions (Figure 3C,D). From this first set of analyses, we can conclude that the predicted interaction of corticosteroids with AQP2 extra-cellular vestibules [18] seems confirmed for cortisol, which, by interacting with the channel extra-cellular mouth in a similar way as for dexamethasone, significantly reduces the pore water permeability. However, in a similar way as for dexamethasone as well, there is a heterogeneity of responses between the four chains.

### 2.3. New Insights on Extra-Cellular C-Loop Regulatory Function: A Focus on Chain A

Before digging deeper into the molecular mechanisms responsible for AQP2 water permeability impairment by cortisol, we focused on the case of chain A to understand its unexpected water free energy profile (Figure 3D). To begin with, we observed that cortisol, indeed, interacted with the extra-cellular vestibule of chain A (Figure 4A). Moreover, the cortisol interacted through hydrogen bonds with residues of the extra-cellular loops of chain A and with the arginine of the ar/R constriction (R187) (Figure 4B). Since the position of the C-loop was shown to correlate with the size of ar/R constriction [18], we investigated the hypothesis of a similar phenomenon occurring, which could explain the unexpectedly low water free energy profile of chain A. First of all, the significant increase in pore diameter at the ar/R constriction (Figure 4C) is in good agreement with the lowering of the chain A water free energy profile as we know the primary role of this constriction in determining the permeability of the channel. Indeed, the ar/R constriction, also called the “selectivity filter”, corresponds to the narrowest part of the channel and has been demonstrated as central in the channel selectivity toward water or other small polar molecules [34,35]. In addition to the composition of the constriction, the position of the conserved arginine (R187 in AQP2) side-chain inside the pore has been associated with a modulation of water permeability [12,33,36]. The increase in ar/R constriction diameter in “COR” is illustrated by Figure 4A,B where we can see the side-chain of R187 in an unusual position, completely folded back against the pore wall instead of pointing inside the pore lumen as in “control”. To estimate the changes in position of the C-loop inside the extra-cellular vestibule, we computed the mean distances between the C-loop and E-loop backbone atoms (Figure 4A–C). While no significant differences could be observed when the whole 200 nanoseconds of trajectory are used for analysis, the distance between the C-loop and E-loop significantly increases in “COR” when the second half of the trajectory only is taken into account (Figure 4C). This can be explained by the time needed for the C-loop conformational change to take place. Similar changes in an intra-cellular loop involved in plant AQPs gating (D-loop) were shown to take place in 15 nanoseconds only [11], even though other studies indicated timescales closer to hundreds of nanoseconds for the same type of conformational changes [37,38]. Altogether, these data suggest one hundred nanoseconds as a plausible timescale for AQP2 C-loop conformational change estimated through C-loop–E-loop mean distance (Figure 4C). We then tested whether the regulatory impact of the C-loop upon water permeability was mediated by the hydrogen bond network existing between the C-loop residues alanine 117 (A117) backbone, asparagine 119 (N119) side-chain and arginine 187 (R187) side-chain. This hypothesis is supported by the number of hydrogen bonds established between A117 and R187 and between N119 and R187 over time (Figure 4D). While in “control”, the hydrogen bond network between the C-loop residues and R187 is maintained over the whole trajectory, in “COR” this network is broken approximately at time t = 50 ns. This is in good agreement with the positional change of R187 side-chain described previously and indicates that 50 nanoseconds were needed for the conformational change in the C-loop necessary for R187 destabilization to happen. In our previous work, we observed the interaction between the R187 side-chain and the corticosteroid as the determinant in order to significantly impair the channel water permeability [18]. In the present situation, cortisol interacts with chain A R187, however, sparsely (Figure 4E), hence, confirming this observation. 

Finally, we tested the hypothesis that C-loop regulation of the channel water permeability could be significantly correlated to the formation of salt bridges between charged residues of the E-loop and the C-loop. However, no significant differences between “control” and “COR” emerged at this time (Figure 4F). Thereafter, it seems that cortisol, through its interaction with residues of the extra-cellular vestibule of chain A, and especially of its C-loop, induced a conformational change responsible for the folding back of the R187 side-chain. This, in turn, significantly modified the size of the ar/R constriction, which, in addition to the probable alteration of the electrostatic profile of the pore, induced the disappearance of water free energy barriers of the extra-cellular half of the conducting pore (see Figure 3). While this alteration is not correlated with a significant increase (or decrease) in water permeability, it offers new insights on the putative regulatory role of the C-loop in human AQPs through its significant impact on R187 side-chain stabilization inside the pore.

### 2.4. Putative Molecular Mechanism of AQP2 Water Channel Permeability Impairment by Cortisol

To understand how cortisol interaction with AQP2 impairs its function, we focused on the two most significantly impacted subunits, chain B and chain D (Figure 3). We observed a significant decrease in the ar/R constriction diameter, however, only of approximately 0.1 angströms, and hypothesized that the main effect on the channel water permeability was mediated through the attenuation of the positive charge of the R187 guanidinium group. It was already well-established through point mutations study that the arginine of the ar/R constriction was the determinant for the selectivity of the channel toward different small polar solutes [35]. Another molecular dynamics study allowed for a better understanding of its role and of the selectivity mechanism of AQPs [34]. Through the potential of mean force calculation, the authors described the selectivity of AQPs as relying on two criteria, both dependent on the composition of the constriction: (i) a steric criterion that discriminates solutes from their size and (ii) a hydrophobic criterion that imposes a free energy barrier at the location of the ar/R constriction induced by protein–water interactions. Indeed, depending on the more or less hydrophobic nature of the constriction, the compensation of water–protein hydrogen bond losses by the solute polar nature is the determinant. The arginine of the constriction is central in this mechanism as it corresponds to the main contribution to the protein–water interaction at this location [34]. Hence, R187 acting directly on both criteria as the position of its side-chain in the lumen of the pore will both modify the narrowest diameter of the conducting pore and its capacity to interact with water molecules [12,18,33,34,36]. In the present study, we investigated which contribution was the most altered by cortisol interaction with the AQP2 extra-cellular vestibule through the case study of the two most impacted chains (Figure 5 and Figure 6). From the pore water free energy profiles of these two chains (Figure 3D and Figure 5A,B), it appears that they were not affected in the same way by cortisol. Indeed, for chain B, the effect of the interaction was concentrated on the ar/R constriction region, while for chain D it spread out over the entire conducting pore.

When looking at the pore diameter (Figure 5C,D), we can see that the sole significant difference between “control” and “COR” for chain B resides in the location of the cortisol molecule, inside the extra-cellular vestibule. On the other hand, for chain D, the diameter of the pore is wider in “COR” than in “control” except at the cortisol fixation site again. As for the pore electrostatic potential profile (Figure 5E,F), in chain B, “control” and “COR” profiles are almost convergent, while in chain D, there is a clear segregation of the two conditions along the entire conducting pore. Altogether, these results suggest that the cortisol impacts water permeability of the channel through the alteration of the hydrophobic selectivity criterion. Indeed, even though there is a significant reduction in the extra-cellular vestibule diameter compared to the “control” condition in both chains, the narrowest part of the channel (i.e., the ar/R constriction) is still as wide (for chain B) or even wider (for chain D) in the “COR” condition. This is in good agreement with our precedent work on the AQP2–dexamethasone interaction where we highlighted that the energetic constriction induced by the fixation of the corticosteroid did not correspond to the steric constriction of the channel [18]. However, this hydrophobic effect is closely localized for chain B at the ar/R constriction site. Confirming this localized effect is the correction of the *pf* with the *Dk* constant (Figure 5G). We introduced the *pf* correction *Dk* constant in another work to tackle an overestimation bias observed on closed channels [12] (see methods). This constant accentuates the effect of the ar/R free energy barrier to better integrate its impact on water permeability. As we can see, when this correction is applied on chain B *pf*, the significant difference between “control” and “COR” conditions correlates with the more straightforward counting approach (Figure 5G). On the other hand, this hydrophobic effect is more global or diffused in the case of chain D as we can see on the free energy profiles (Figure 5B). Moreover, the pore diameter and pore electrostatic profiles (Figure 5D–F) indicate a probable reorganization of pore lining residues.

In Figure 6A,B, we can see that while the cortisol formed hydrogen bonds with the R187 side-chain over the entire 200 nanoseconds of trajectory for chain D, it was not the case for chain B where the interaction was maintained during half of the trajectory only. Chain D was also the most impacted subunit by the interaction with cortisol (Figure 3), which correlates with the most negative binding free energy as well (Figure 6C,D). Hence, dexamethasone and cortisol seem to modulate water permeability of human AQP2 in a similar manner [18]. As the interaction between cortisol and the R187 side-chain was not as stable in chain B as in chain D, we hypothesized that the more diffuse hydrophobic effect visible on chain D could be a consequence of the attenuation of the R187 positive charge, which, in turn, would have led to a reorganization of the water interaction sites. It is well-known that AQPs water transport is mediated by a succession of polar water interaction sites opposed to hydrophobic residue side-chains, which are responsible for the typical single file continuum of water molecules inside the pore [3,39]. Most of these water interaction sites are constituted by the oxygen atoms of the backbone of loops B and E [40]. Hence, to estimate the impact of cortisol interaction on the water interaction site network, we used the angle formed between these residues’ backbone atoms C and O and the z axis (Figure 6E–H). As expected, significant differences between “control” and “COR” conditions are visible for almost all interacting sites in chain D, contrary to chain B. Figure 6I,J illustrates these impacts on the electrostatic potential profiles of the conducting pore.

In the center of the channel, the dipole moments of hemi helices HB and HE terminating at the NPA motifs irradiate the pore positively. This is in good accordance with the well-documented proton exclusion mechanism of AQPs [41]. The negative zones of the backbone oxygens of the water interaction sites are also visible. However, we can clearly see the attenuation of the positive guanidinium group of R187 in chain D and the resulting accentuation of the electronegativity of its water interaction sites. To sum up, in a similar manner as dexamethasone [18], cortisol impacts AQP2 water permeability mainly through the modification of the hydrophobicity of the pore (Figure 5). This can affect the corticosteroid interaction site—the extra-cellular vestibule—only as in chain B (Figure 5) or the whole conducting pore when the cortisol interacts directly with the side-chain of the arginine of the ar/R constriction (Figure 6). This can be explained by the attenuation of the positive charge of the arginine, which modifies the whole pore electrostatic profile (Figure 5F) and induces a reorganization of the water interaction site network (Figure 6).

### 2.5. Spontaneous Binding of Cortisol with Human AQP2 Extra-Cellular Vestibules

To comfort the likelihood of the interaction, a more realistic atomic system was built mimicking the intra-cellular and the extra-cellular compartment (Figure 7G) through the use of different cations (sodium ions were placed in the extra-cellular space and potassium ions in the intra-cellular compartment, see methods). In the beginning of the simulation, four cortisol molecules were placed in solution in both compartments. Interestingly, the cortisol molecules behaved differently depending on the compartment (see movie). In the intra-cellular compartment, cortisol barely interacted with the protein and three of them solubilized into the POPC bilayer (Appendix A). In the extra-cellular compartment, however, two cortisol molecules interacted spontaneously (without any external forces applied) with the protein for at least 150 nanoseconds consecutively (Appendix A). One of them reached the predicted corticosteroids interaction site and formed a hydrogen bond with the arginine of the ar/R constriction in approximately 20 nanoseconds only (Figure 7E). This interaction was then stabilized over the quasi totality of the 250 nanoseconds of trajectory. The mean binding free energy of this interaction is equal to −9.39 Kcal·mol^−1^ (Figure 7F), which corresponds to a K_D_ of 239.17 nM. This K_D_ corresponds to specific interactions [42] and is close to what we obtained for AQP2–dexamethasone interaction [18]. An illustration of the tetrameric assembly of AQP2 at the end of the simulation (time t = 250 ns) shows two cortisol molecules docked into chain B and chain C (Figure 7H). The number of hydrogen bonds between one cortisol and chain B R187 shows that this interaction is a recurrent event, again supporting its likelihood (Figure 7E). Finally, when we compared the water permeability of chain B with the one of chain D, for which there was no interaction with cortisol, a net significant difference was obtained (Figure 7B). This difference is also illustrated by the cumulative number of water permeation events along the simulation time with a very stable plateau phase visible for chain B (Figure 7A). These results point toward a regulation of human AQP2 function by cortisol in physiological conditions and are very similar to the results obtained for the AQP2–dexamethasone interaction [18]. The repeatability of this spontaneous interaction of corticosteroids with the extra-cellular vestibules of AQP2 is in good agreement with a real biological regulatory function.

### 2.6. Additional Insights Supporting the Regulatory Function of Ions Nature and of C-Loop

In our precedent work, we also noticed the impact of cation nature on the function of AQP2. Indeed, sodium ions, because of both their smaller van der Walls radius and higher electropositivity, were “stickier” than potassium ions with the carboxylates of the protein’s surface. This resulted in the disruption of salt bridges between the extra-cellular C-loop and E-loop and allowed for a rearrangement of the C-loop, which, in turn, modified the channel water permeability through the repositioning of the R187 side-chain (see previous sections) [18]. This phenomenon was observed again in the present study for chain A (in experimental setup 2, see methods). This chain did not interact with cortisol during the 250 nanoseconds of trajectory but still displayed a significant difference with chain D for water permeability (Figure 7A,B). The radial distribution functions of sodium ions with the carboxylates of the extra-cellular surface of chain A or chain D showed a clear over-accumulation of the cations within approximately 2 angströms of the carboxylates for chain A (Figure 7C). Additionally, the mean distance between the E-loop and C-loop backbone atoms indicates a rearrangement of the chain A and chain B C-loop compared to chain D (Figure 7D). The regulatory potential of ions upon AQPs function was already investigated through the impact of voltage or membrane potential on their water permeability; however, no clear experimental proof allowed for a robust conclusion in favor of a physiological relevance of such regulation [12,36]. This result, together with our previous study [18], seems to indicate the nature of ions to be more determinant than their concentration or than membrane potential in regulating AQPs water permeability. It also strengthens the regulatory relevance of charged residues of the extra-cellular surface of AQPs already pointed out by several molecular dynamics studies [12,43].

The correlation between E-loop–C-loop mean distance and the permeability of the channel (Figure 7B–D), together with the particular case of a larger destabilization of R187 in the first system chain A example (see previous sections, Figure 4), points toward a regulatory role of the extra-cellular C-loop. It also highlights the crucial role of the hydrogen bonds network formed by these residues in stabilizing the arginine of the ar/R constriction and, hence, in modulating the channel water permeability. This observation is also supported by a recent molecular dynamics study correlating the propensity of the arginine of the ar/R constriction to change conformation to the number of hydrogen bonds stabilizing it and to the permeability of the channel [44]. However, further studies need to be carried out to validate these observations and to decipher the intricacies of the interactions of the extra-cellular C-loop with all types of biomolecules and ions.

## 3. Discussion

### 3.1. Cortisol Is a Putative Ligand of AQP2

In the present study, through molecular dynamics, we observed a significant impact of cortisol on the water permeability of AQP2. We described the molecular mechanism underlying such water transport inhibition: through its direct interaction with the residues of the extra-cellular vestibule of AQP2 (in particular with R187 of the ar/R constriction), cortisol modifies the electrostatic potential of the conducting pore, which in turn modifies the position of protein–water interaction sites and compromises the entire hydrogen bonds network. Moreover, this interaction happened spontaneously in a more realistic atomic system where the cortisol molecules were placed in solution and corresponded to a mean K_D_ of 239.17 nM. Adding up to the recently published similar effect of dexamethasone upon AQP2 function [18], these results indicate the corticosteroids family as putative regulators of transmembrane water fluxes through their direct interaction with AQPs extra-cellular surface. Both dexamethasone and cortisol inhibit water fluxes the most significantly by forming hydrogen bonds with the very conserved arginine of the ar/R constriction (R187 in AQP2). Moreover, this interaction occurs spontaneously and is stably maintained over several hundreds of nanoseconds for both molecules. Therefore, based on this criteria and on similarities between human AQPs extra-cellular vestibules and the binding site of MR and GR and the CCM, we defined a first AQPs corticosteroid binding site (ACBS).

Molecular dynamics simulations of steroids in a POPC environment have studied the mechanisms of steroid insertion in membranes [45]. These studies have concluded that steroids can insert quickly and cross membranes, with the flip-flop events inside the membrane constituting the limiting step. They observed that steroids spontaneously enter the membrane at rates ranging between 10^−1^·s^−1^ for planar membranes with pronounced unstirred layers and 10^5^·s^−1^ for large unilamellar vesicles. During the 250 ns simulated, three cortisol molecules entered the membrane (see movie), which corresponds to two orders of magnitude above the expected higher entry rate of 10^5^·s^−1^. This very quick integration of cortisol into the membrane could be due to the interaction of cortisol with the AQP2 dipole moment. Indeed, the three cortisol penetrating the membrane were all located into the intra-cellular compartment. On the other hand, three cortisol of the extra-cellular compartment interacted with AQP2 (Figure 7 and Appendix A). In a previous work, we observed a significant correlation of dipole moments of dexamethasone with AQP2 [18]. In a similar manner, this type of long distance interaction could be held accountable for an attraction of the steroids toward AQP2 extra-cellular vestibules and for a repulsion in the intra-cellular compartment, accelerating, in turn, the entry of cortisol into the lowest lipid bilayer (see movie). Moreover, the equally high rate of interaction of cortisol with the AQP2 extra-cellular surface (3 cortisol/250 ns or 12.10^6^·s^−1^) indicates a higher affinity of this steroid for the ACBS than for the POPC membrane [45]. Finally, the three cortisol that entered the membrane were mostly disposed horizontally, which is in good agreement with the observations of Atkovska et al., supporting the reliability of our results [45].

### 3.2. Implications for Steroid Non-Genomic Effects

Evidence for the existence of non-genomic effects of corticosteroids mediated by a novel membrane corticosteroid receptor is increasing [28,29,46,47,48,49]. The transduction pathway of the non-genomic effects of corticosteroids is still under investigation, and, though there is no clear picture depicted at present [28], TPR channels have been particularly involved (reviewed in [50,51,52]. TRPV1 and TRPV4 channels are mechanosensitive calcium channels from the transient receptor potential vanilloid family (TRPV) characterized as facilitating calcium influx into the cell when the plasma membrane is stretched [53,54]. Though some steroids have been shown to be ligands of specific TRP channels by molecular docking simulations [52], there is scarce experimental evidence, except for testosterone and the TRPM8 receptor [55]. Strikingly, several studies have pointed out functional and physical interactions between AQPs and TRPV1 and TRPV4 [56,57,58,59,60]. Altogether, these data suggest the AQP–TRPV complexes as realistic and efficient non-genomic signal transduction initiators. Coupled to a direct regulation of water fluxes by corticosteroids interaction with AQPs, these transmembrane proteins could participate in the integration of the rapid non-genomic effects of corticosteroids, depending on the corticosteroid: considering the AQP couple, their interaction could lead to inhibition, and as for AQP2, glucocorticoids such as dexamethasone [18] or cortisol (see results), or the activation of water transport through the membrane. The resulting local stretching or compression of the membrane would trigger the opening or the closure of TRPV channel co-localized near the AQP, leading to sudden changes in intra-cellular calcium concentration ([Ca^2+^]_i_). AQP2 is expressed in many organs of the human body [61] but is most known for its role in vasopressin—mediated water re-absorption in the kidneys [62,63]. Expressed in principal cells of the renal collecting duct, AQP2 is the primary target for short-term regulation of collecting duct water permeability and is regulated by vasopressin through the modulation of its cellular localization [63]. Corticosteroids also participate in maintaining body water and salt homeostasis through their action on the kidneys’ tissues. In the renal distal tubules and collecting duct, mineralocorticoid (aldosterone or cortisone)/glucocorticoid (cortisol) balance determines water, sodium and potassium re-absorption. Indeed, mineralocorticoids (aldosterone) are well-known for triggering sodium and water uptake coupled with potassium loss [64], while glucocorticoids (cortisol) are responsible for water clearance and have been reported to have an antagonist effect with vasopressin [65,66,67]. So far, we have shown an inhibitory effect of two glucocorticoids (dexamethasone and cortisol) upon AQP2 water transport. Based on their antagonist effect in the kidneys, one could hypothesize an opposite regulation of AQP2 activity segregating mineralocorticoids and glucocorticoids. Hence, mineralocorticoids should be associated with open conformation of AQP2 and water influx and [Ca^2+^]_i_ increase, while glucocorticoids would diminish water influx and prevent [Ca^2+^]_i_ increase. In good agreement with this hypothesis, a study from Grossmann et al. highlighted a net [Ca^2+^]_i_ increase triggered by aldosterone application but not by dexamethasone application on cultured cells [68]. However, up to this day, the data are too scarce to conclude in favor of AQPs as membrane corticosteroid receptors for non-genomic actions even though they constitute, in association with TRPV1 and TRPV4 calcium channels, coherent candidates. Many interrogations remain, starting with the mean predicted K_D_ of 239.17 nM, which is close to some experimental values for glucocorticoid membrane receptors (180 nM obtained by fluorescence correlation spectroscopy [69]) but very far from others (from 0.16 nM to 15 nM [70,71,72,73,74,75]). These divergences could be linked to differences in the ACBS of AQPs or to the existence of other membrane proteins, such as ion channels targeted by corticosteroids as well. Moreover, many of the described non-genomic actions of corticosteroids seem associated with G-protein coupled receptors (GPCR) [70,71,72,73,75,76]. Hence, G-protein could be directly regulated by corticosteroids through interactions with GPCR or indirectly through the existing dependencies between G-proteins, cell volume regulations and Ca^2+^ fluxes [77,78]. Finally, the investigation of mineralocorticoids interaction with AQPs, with regards to the inhibitory effect described here for glucocorticoids, appears to us as of primary interest. 

As a conclusion, this study provides the framework for investigating AQPs as potential receptors of corticosteroids. The in silico analysis provides some evidence of a specific interaction. Future in vitro and in vivo investigations will be needed to confirm this hypothesis and evaluate if AQPs should be considered as new actors regarding the non-genomic effects of corticosteroids.

## 4. Material and Methods

### 4.1. Molecular Dynamics Simulations

All simulations were performed with Gromacs (v.2018.1) [79] in a CHARMM36m force field [80]. The systems were built with the CHARMM-GUI interface [81,82]. A first minimization step was followed by six equilibration steps, during which, restraints applied on the protein backbone (BB: N CA C O), side chains (SC: side chains heavy atoms) and on lipids (LIPID: polar head heavy atoms) were progressively removed (energies are given in KJ/mol/nm^2^. Phase 1: BB = 4000.0, SC = 2000.0, LIPID = 1000.0; Phase 2: BB = 2000.0, SC = 1000.0, LIPID = 400.0; Phase 3: BB = 1000.0, SC = 500.0, LIPID = 400.0; Phase 4: BB = 500.0, SC = 200.0, LIPID = 200.0; Phase 5: BB = 200.0, SC = 50.0, LIPID = 40.0; Phase 6: BB = 50.0, SC = 0.0, LIPID = 0.0) before the production phase was performed without restraint. Pressure and temperature were kept constant at 1 bar and 310.15 Kelvin, respectively, using the Berendsen method during equilibration and the Parrinello–Rahman and Nosé–Hoover methods during production. The Lennard-Jones interaction threshold was set at 12 Angströms (Å) and the long-range electrostatic interactions were calculated through the particle mesh Ewald method.

Three experimental setups were carried out.

Experimental setup 1 corresponds to Figure 3, Figure 4, Figure 5 and Figure 6: The tetrameric assembly of AQP2 (pdb: 4nef) was inserted into the POPC bilayer, solvated with TIP3 water molecules and 150 mM of KCl for “control” conditions. For the “COR” condition, additionally, 4 cortisol molecules were manually placed inside the extra-cellular vestibules of AQP2 following the same procedure as in [18], hence, leading to one cortisol per monomer. The two systems were then simulated for 200 nanoseconds.

Experimental setup 2 corresponds to Figure 7. The same starting tetrameric assembly as for the setup 1 condition “control” (i.e., without the cortisol fixed inside the extra-cellular vestibules) is used. The tetramer is then inserted into the POPC bilayer and solvated with TIP3 water molecules. An additional POPC bilayer without AQP was fused to the initial system to compartmentalize it. Then, in the extra-cellular compartment, 150 mM of NaCl was added, and in the intra-cellular compartment, 150 mM of KCl was added. To mimic a membrane potential, an additional 1.25 ion pair was placed asymmetrically (one ion pair and one pair of modified ions with charge equal to +0.25 and −0.25). The Na+ ions were placed in the extracellular compartment and the Cl- ions in the intra-cellular compartment [12,18,36]. As a result, a membrane potential of 40 mV was obtained. Finally, four molecules of cortisol were solubilized into the extracellular compartment and four others were solubilized into the intra-cellular compartment. The system was then simulated for 250 nanoseconds.

Experimental setup 3 corresponds to Figure 1 and Figure 2. For Figure 1, the experimental setup 1 condition “COR” was used. Additionally, another system was built with dexamethasone instead of the cortisol molecules for the “DEX” condition. This system was then simulated for 200 nanoseconds. For Figure 2, a portion of trajectory from experimental setup 2 at a length of 50 ns from time t_0_ = 50 ns to time t_e_ = 100 ns was used. The section was chosen based on the closed state of the channel as well as on the continuous interaction between chain B R187 and the cortisol (Figure 7A–E).

### 4.2. Analysis

#### 4.2.1. Water Permeability

To monitor water molecule displacement along the trajectories, the MDAnalysis library is used [83,84]. From these water coordinates, the water count and permeability coefficient (*pf*) are derived. Permeability coefficients were calculated according to the collective coordinate method [85]. In Figure 5, based on the free energy profiles, a correction is applied to *pf* to integrate better free energy barriers [12,18]. This correction constant is calculated as follows: $Dk=\left[2E_{0}-E_{\left(arR\right)}\right]/E_{0}$ with *Dk* as the unit free correction constant; *EarR* as the free-energy at the ar/R constriction site and *E*_0_ as the free energy corresponding to the highest free-energy barrier in the channel section used to calculate *pf* (indicated in Figure 5A). To adjust the *pf*, one has to multiply it by *Dk*: *pf_corrected_ = pf* × *Dk*

#### 4.2.2. Free Energy Profiles

Water free energy profiles were extrapolated from the logarithm function of the water counts inside the pore with the z-axis as a reaction coordinate [33,34]. The pore is divided along the reaction coordinate (z axis) in slices of 0.5 Angströms (Å). The average density of water molecules in each slice is then computed over the 250 ns of simulation, and the Gibbs free energy *G*(*z*) is obtained as follows:$$G\left(z\right)=-KT\ln\frac{\rho \left(z\right)}{\rho_{bulk}}$$
where *K*, *ρ_bulk_* and *T* represent the Boltzmann constant, the bulk density and the absolute temperature, respectively.

#### 4.2.3. Binding Free Energy and Dissociation Constant

The binding affinity of cortisol to AQP2 was evaluated directly from the structure, extracted from the molecular dynamics trajectories every nanosecond with the PRODIGY-LIG program [86]. PRODIGY-LIG evaluates the contacts between ligand and protein and computes a free binding energy from a reliable empirical equation [87].

Dissociation constant (*K_D_*) values were obtained from the binding free energies as follows:$$K_{D}=\exp(\frac{-\Delta G_{S}}{RT})$$
with Δ*G_S_*, *R* and *T* as the binding free energy, the perfect gaz constant and the temperature, respectively.

#### 4.2.4. Other Properties

Membrane potential, hydrogen bonds, distances and radial distribution functions (rdf) were computed with GROMACS tools (version 2020.6). Pore profiles were computed with HOLE software [88].

#### 4.2.5. Statistical Analysis

All statistical analyses were performed using R programming language. Before any statistical test was performed, normality and homoscedasticity of the variables were controlled to choose between parametric or non-parametric tests. When two variables were compared, the Student *T* test or Mann–Whitney test was used. When more than two variables were compared, the Tukey post hoc test after one-way analysis of variance or Bonferroni post hoc correction after the Wilcoxon test was used.

For experimental setup 1, the 200 ns trajectories were divided into 10 ns sub-trajectories and the analyses were performed for each monomer, yielding 80 repetitions per condition.

For experimental setup 2, the 250 ns trajectories were divided into 10 ns sub-trajectories and the monomers were compared between each other, yielding 25 repetitions per condition.

#### 4.2.6. Structural Alignment of Experimentally Solved Structures and Delineation of a Putative Conserved Binding Site to Corticoids

The superposition was achieved for each structure on the AQP2 structure with the “iterative magic fit” routine of the Swiss PdbViewer-4.1 software [89] by selecting “all atoms”.

The ligand (COR) was previously placed in a bound state from the AQP2-COR dynamics by superposition on the AQP2 structure to determine candidate residues to be a corticoid binding site. A 50 ns trajectory segment (50 to 100 ns from experimental setup 2) displaying a docked subunit (chain B) with COR was used to grasp the effective residues in retaining the cortisol in a bound state.

## Figures and Tables

**Figure 1 ijms-24-01499-f001:**
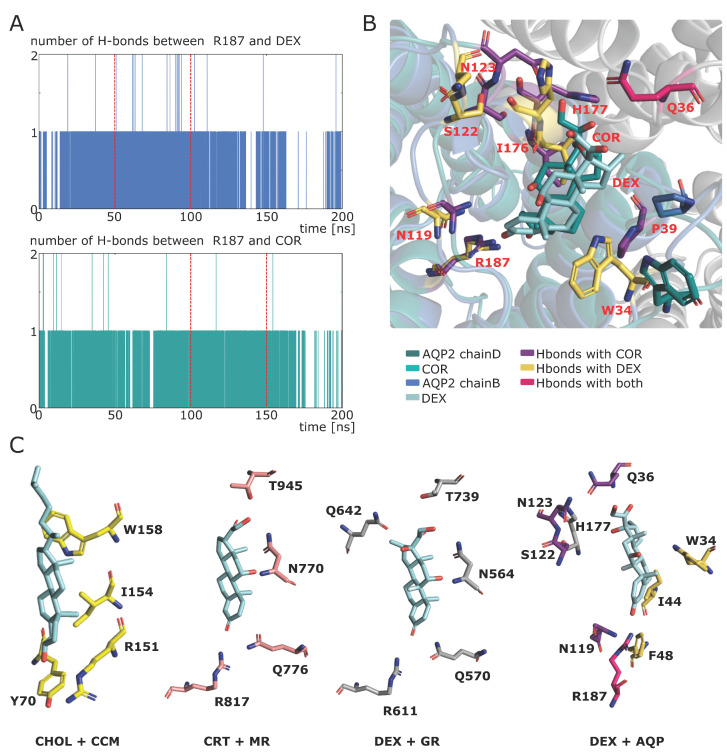
Functional homologies of Cholesterol Consensus Motif (CCM), Mineralocorticoid Receptor (MR) and Glucocorticoid Receptor with AQP2 extra-cellular vestibule. (**A**) Number of hydrogen bonds established between AQP2 chain B and dexamethasone (in blue) and between AQP2 chain D and cortisol (in green). The portion of trajectory used to compute corticosteroid binding site hydrogen bond contributors are indicated by red dashed lines. (**B**) Schematic representation of AQP2 chain B with dexamethasone and of AQP2 chain D with cortisol. All residues establishing hydrogen bonds with the corticosteroid during the 50 ns of simulation indicated on part A are colored in yellow (for dexamethasone interaction) and in purple (for cortisol interaction). Glutamine 36 of the adjacent chain is colored in pink and interacts with the corticosteroid in both cases. Name of residues are indicated in red. (**C**) Schematic representation of cholesterol consensus motif and several corticosteroid receptors binding site. CHOL + CCM: cholesterol in the cholesterol consensus motif of human β2-Adrenergic receptor (pdb: 3D4S). CRT + MR: corticosterone in interaction with human mineralocorticoid receptor (pdb: 2A3I). The residues involved in hydrogen bonds with the corticosteroid are represented. DEX + GR: dexamethasone in interaction with human glucocorticoid receptor (pdb: 1M2Z). The residues involved in hydrogen bonds with the corticosteroid are represented. DEX + AQP: dexamethasone in interaction with human AQP2 (pdb: 4NEF). Residues involved in hydrogen bonds with the corticosteroid and with a similar nature and position as in MR and GR are colored in purple. Residues mimicking cholesterol consensus motif are colored in yellow. The conserved ar/R R187 involved in both is colored in pink. H177 is a residue involved in hydrogen bonds different from MR or GR residues which could act as a pH sensor.

**Figure 2 ijms-24-01499-f002:**
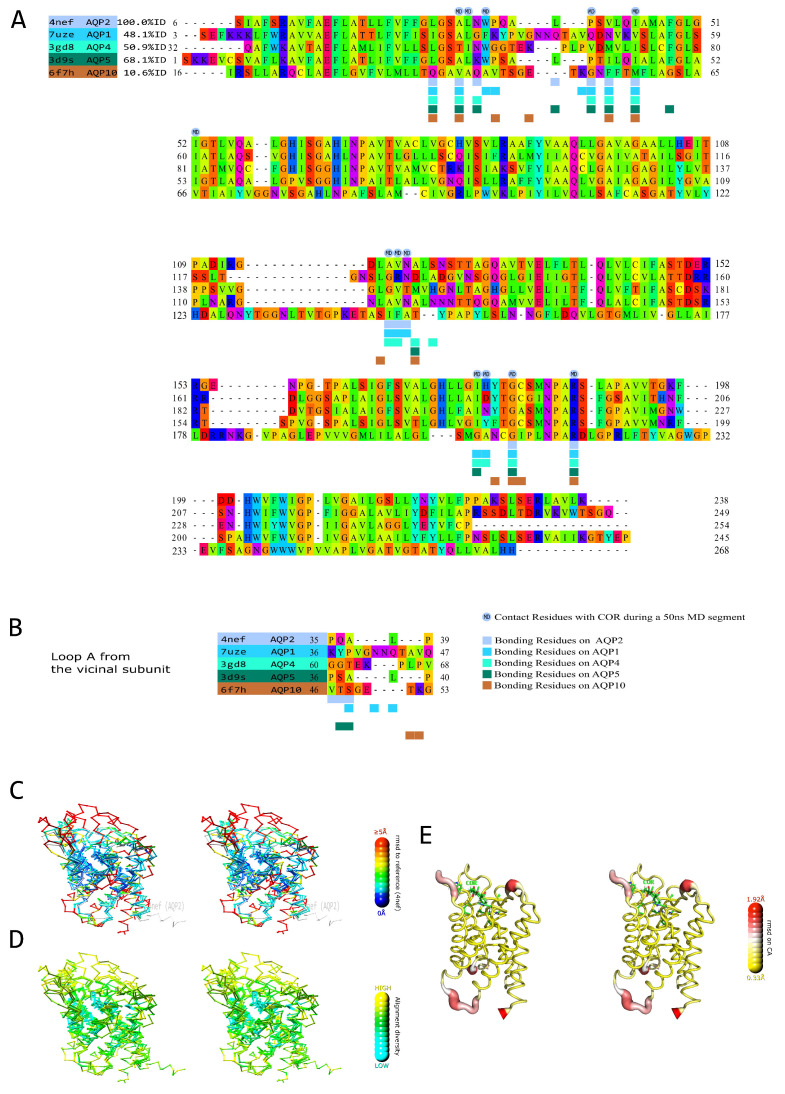
Structural alignment of experimentally solved structures of 4 different human AQP representatives against the structure of human AQP2. The PDB codes of the corresponding structures are recalled at the beginning of the alignment. The template to color the residues (Taylor Scheme) is meant to highlight the conversation of the physical–chemical properties. (**A**) Structural alignment of the binding subunit. The contact points are deduced out of the solved conformations and marked by a square colored according to the sequence label color. Above the alignment, a light-blue circle indicates the effective contacts during a 50 ns bound state (to the central arginine) trajectory segment. (**B**) Contact points on the adjacent subunit with the bound ligand (Domain Swapping example). (**C**) Stereo view of the relative structural superimposition displaying the rms distance (root mean square) with the template in a blue to red rainbow gradient (output from Swiss PdbViewer-4.1). The used scale is the default one that is linear where the dark blue is for rms = 0 Å, and red is for rms >= 5 Å. (**D**) Stereo view of the relative structural superimposition displaying the alignment diversity in a cyan–green–yellow gradient (output from Swiss PdbViewer-4.1) to all layers, which purposed is to display the degree of similarity among all aligned residues. **E.** Stereo view of a “putty representation” of the 50 ns trajectory segment in a yellow–white–red gradient (output from PyMOL). The residues implicated in the docking of the ligand are shown in licorice (distance < 4 Å). RMSD on CA atoms ranges between 0.33 and 1.92 Å.

**Figure 3 ijms-24-01499-f003:**
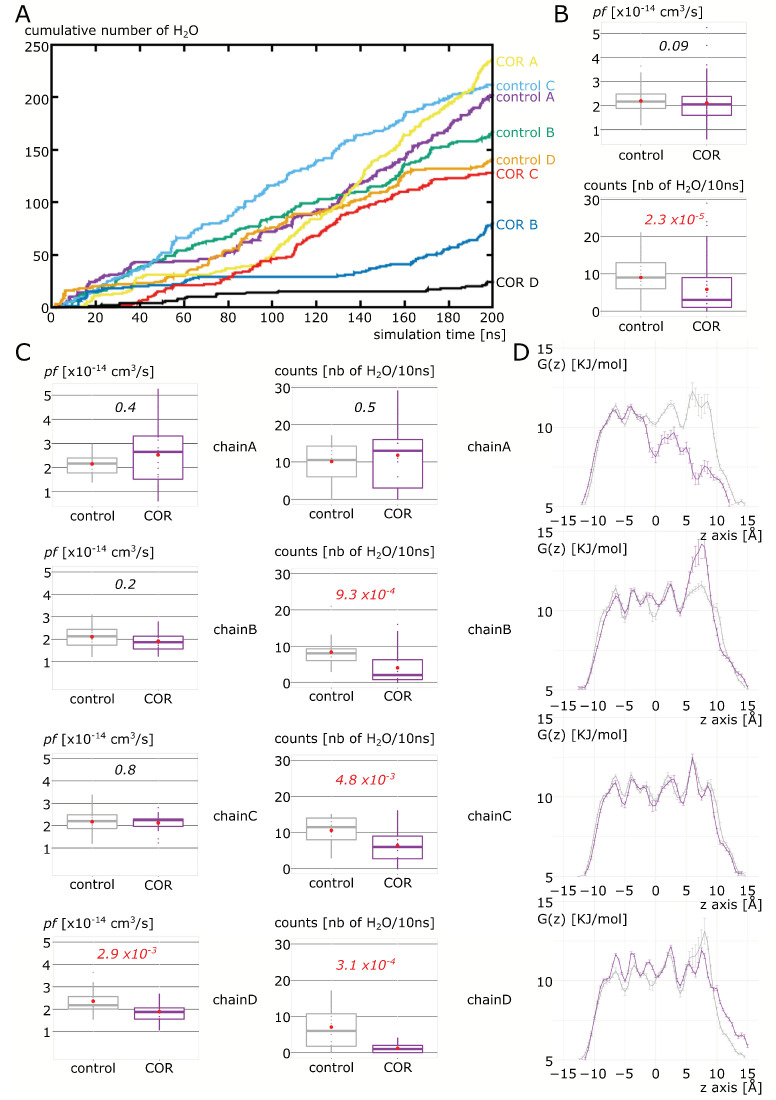
Impact of cortisol on AQP2 water permeability. (**A**) Cumulative number of water crossing the whole transmembrane pore section of 30 angströms as a function of simulation time. Both conditions “control” and “COR” are represented. (**B**) Osmotic permeability coefficient (pf) and number of water crossing the entire 30 angströms-long pore section compared between “control” and “COR” conditions. (**C**) Same water permeability indicators between the two conditions for each monomer (chain A, chain B, chain C and chain D). For statistical comparisons, non parametric Mann–Whitney test was performed, and *p* values are indicated in italic and colored in red when corresponding to alpha < 0.05. (**D**) Water free energy profiles centered on the center of the conducting pore and zoomed over the 30 angströms-long transmembrane section used to count water permeations for each monomer. “Control” condition is represented in gray and “COR” condition in purple.

**Figure 4 ijms-24-01499-f004:**
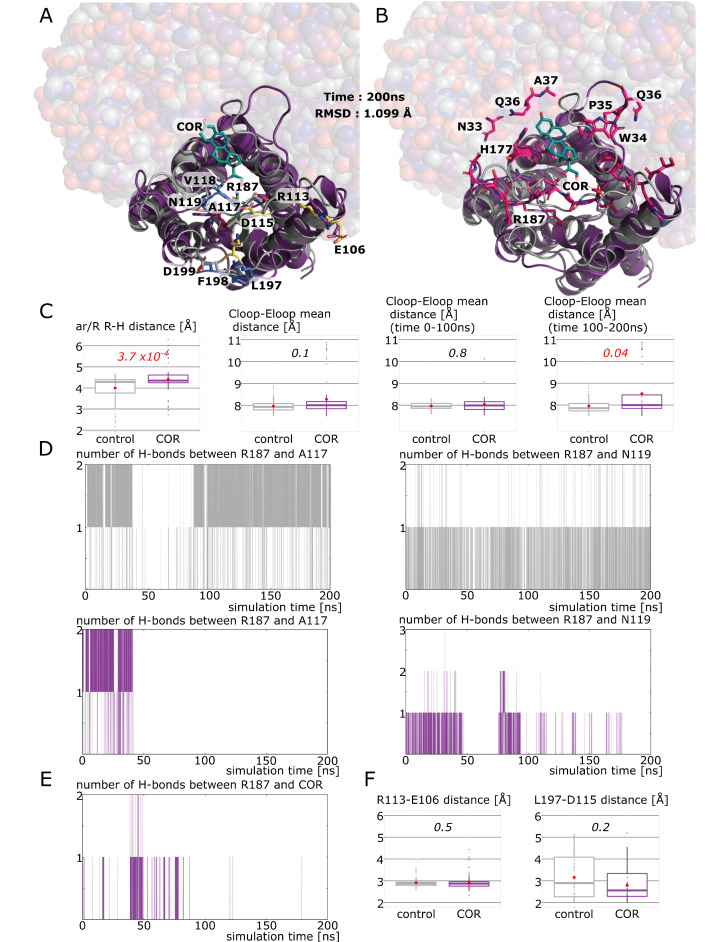
New insights on extra-cellular C-loop regulatory function. (**A**,**B**) Representation of AQP2 chain A at time = 200 ns. The three other chains are represented with spheres. “Control” condition is colored in gray and “COR” condition in purple. Cortisol in interaction with chain A is represented in green. (**A**) Arginine of the ar/R constriction is colored in green in “control” and in pink in “COR”. Residues used for C-loop–E-loop mean distance calculations are colored in blue, and residues used for R113–E106 and L197–D115 smallest distance calculations are colored in yellow. (**B**) All residues establishing hydrogen bonds with the cortisol in “COR” during the 200 ns of simulation are colored in pink. (**C**) From left to right: smallest distance between the arginine and the histidine of the ar/R constriction; C-loop–E-loop mean distance (alpha carbon of residues colored in blue in A are used) over the entire trajectory; C-loop–E-loop mean distance over the first half of the trajectory (0 ns to 100 ns) and over the second half of the trajectory (100 ns to 200 ns). Conditions were compared with non-parametric Mann–Whitney test and *p* values are indicated in italic. (**D**) Number of hydrogen bonds between the arginine of the ar/R constriction (R187) and two other residues of C-loop (A117 and N119) as a function of simulation time. “Control” condition is in gray and “COR” in purple. (**E**) Number of hydrogen bonds between R187 and cortisol in “COR” condition as a function of simulation time. (**F**) Smallest distances between C-loop and E-loop residues involved in salt bridges. Mann–Whitney test was used to compare conditions. No significant differences between conditions appeared.

**Figure 5 ijms-24-01499-f005:**
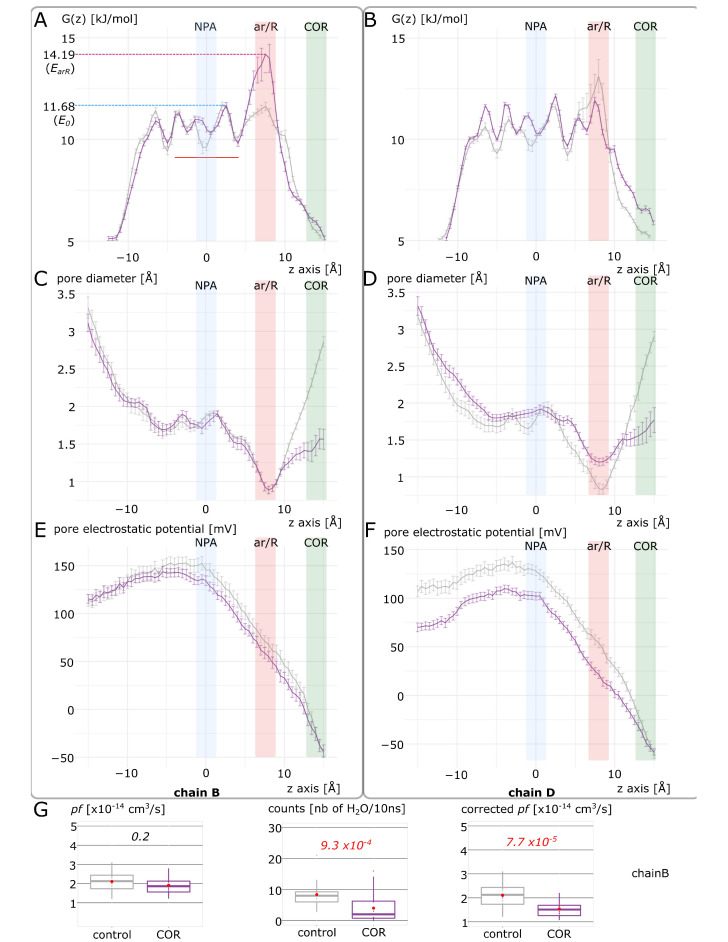
Differences in electrostatic and steric contributions to water permeability hindrance by cortisol. “Control” condition (in gray) is compared to “COR” condition (in purple) for chain B (**A**,**C**,**E**) and chain D (**B**,**D**,**F**). (**A**,**B**) Water free energy profiles inside the pore. (**C**,**D**) Pore diameter. (**E**,**F**) Pore electrostatic potential. The NPA motifs corresponding to the center of the channel are indicated in blue, the ar/R constriction in red and the extra-cellular vestibule where the cortisol is docked to the porine in green. (**G**) Estimation of ar/R constriction contribution to water permeability hindrance in chain B through Dk constant pf correction. From left to right: pf, number of water permeation events (crossing the entire membrane section), corrected pf. The two conditions were compared with non-parametric Mann–Whitney test for pf and counts and with parametric *t*-test for corrected pf. *p* values are indicated in italic.

**Figure 6 ijms-24-01499-f006:**
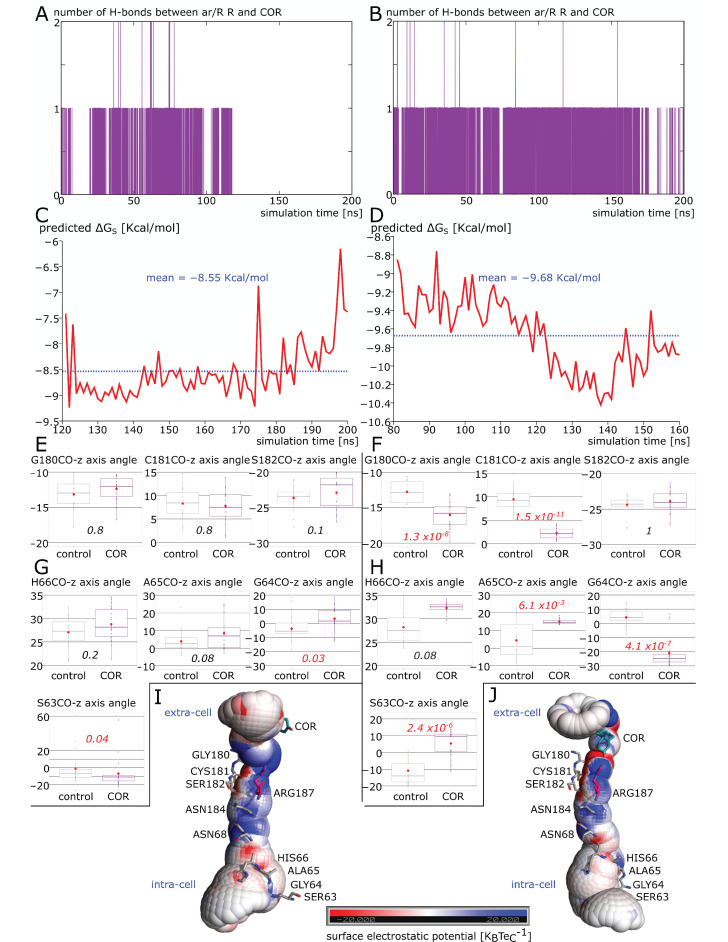
Impact of cortisol interaction with ar/R arginine upon AQP2 pore structure and electrostatic potential profile. In this figure, data corresponding to chain B are always displayed on the left while data corresponding to chain D are on the right. (**A**,**B**) Number of hydrogen bonds between cortisol and ar/R constriction arginine 187 as a function of simulation time for chain B and D, respectively. (**C**,**D**)**.** Binding free energy of cortisol with extra-cellular vestibule of chain B and D, respectively. (**E**–**H**) Comparison between conditions “control” and “COR” of the angle formed between C–O axis of pore lining residues and the z axis. Positive values indicate oxygen atoms orientated toward the extra-cellular extremity of AQP2 and negative values correspond to the opposite. E and F display data for pore lining residues of the extra-cellular half of AQP2, and in G and H, the ones corresponding to the intra-cellular half. Conditions were compared with non-parametric Mann–Whitney test and *p* values are indicated in italic. (**I**,**J**) Schematic representation of the conducting pore on which has been projected the electrostatic surface potential (±20 K_B_Te_C_^−1^). Pore lining residues (in gray), NPA motifs asparagines (in gray), ar/R constriction arginine (in pink) and cortisol (in green) are represented. Cellular compartments are indicated in blue.

**Figure 7 ijms-24-01499-f007:**
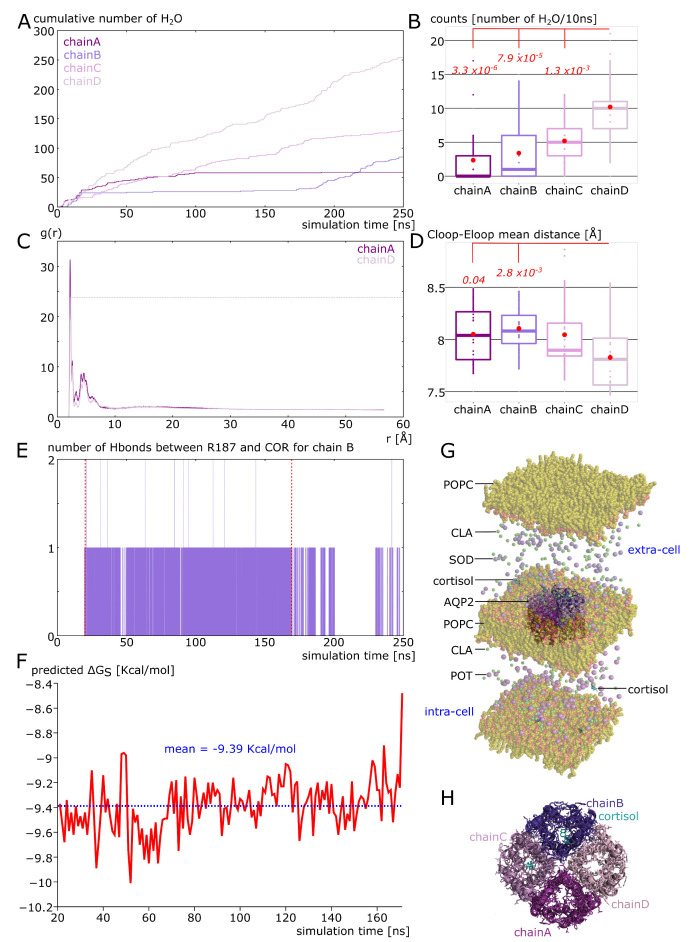
Spontaneous interaction of cortisol with AQP2 extra-cellular surface. (**A**) Cumulative number of water molecules crossing the whole transmembrane section in the pore of chain A, B, C or D along simulation time. (**B**) Comparison of the number of water molecules crossing the conducting pore between the four chains of AQP2. The significant differences with chain D-associated *p* values are indicated in red and are issued from a Bonferroni post hoc correction after Wilcoxon test. (**C**) Radial distribution function of sodium ions with the extra-cellular surface carboxylates (Glu106, Asp111, Asp115, Asp199 and Asp200) of chain A or chain D as a reference. (**D**) C-loop (α carbons of residues Lys197, Phe198 and Asp199)—Eloop (Ala117, Val118 and Asn119) mean distance. The significant differences with chain D-associated *p* values are indicated in red and are issued from a Bonferroni post hoc correction after Wilcoxon test. (**E**) Number of hydrogen bonds between the arginine of the ar/R constriction of chain B and cortisol along simulation time. (**F**) Predicted binding free energy of cortisol with chain B extra-cellular vestibule from time 20 ns to time 170 ns (indicated by red dashed lines in part **E**). (**G**) Schematic representation of the atomic system simulated. (**H**) Schematic representation of AQP2 tetramer as seen from the extra-cellular compartment at the end of the simulation at time 250 ns.

## Data Availability

The data presented in this study are available on request from the corresponding author. The data are not publicly available due to the lack of a dedicated database.

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
