# Peer review of "Cortisol Interaction with Aquaporin-2 Modulates Its Water Permeability: Perspectives for Non-Genomic Effects of Corticosteroids"

_ijms, 2023, doi:10.3390/ijms24021499_

Round 1

Reviewer 1 Report

In the present work, the authors studied the AQP family as a putative corticosteroids membrane receptor. They evaluated the impact of cortisol interaction with human AQP2 and explained the molecular consequences on the water channel properties. The authors have made a careful exposition of the molecular modeling and molecular dynamics simulations methods and data analysis techniques. Spell check is required; for example in Page no. 3: Figure 1 legend: "The residues involved in hudrogen bonds with the corticosteroid are represented." Overall this is an interesting manuscript showing naturally produced glucocorticoid cortisol might inhibit channel water permeability and presenting the putative underlying molecular mechanism. 

Author Response

The corrections suggested have been made. The text has been revised to remove the eventual typos. Track-changes can be displayed in the word document.

We want to thank reviewer 1 for her/his comment that helped to correct our article.

Reviewer 2 Report

This is interesting results,

however, the text does not have lines numebre what does not allow to write full review to it in the current form. Figure 1 is missing as well.

Some general points:

Abstract, line 8: “non-genomic effects” - despite such definition has been accepted, it is better to reformulate it. Epigenetic (rapid histone modification) or protein modofication (phosporylation, for example).

Introduction: “drugs against AQPs” maybe it is better to write “against AQP action”?

Please, reload text with lines number and add figure 1 at least.

Author Response

All apologies for the missing figures, this was independent of our will. The corrections suggested have been made.

We do not agree with the sense of the term “non-genomic effect” as understood by reviewer 2 and maintain its use in this particular case. In fact, the concepts of “non-genomic effects” and the “epigenetic modifications” are different since the mechanisms that are induced are not of the same nature:

_ the time scales involved are not the same (while the “non-genomic effects” can be considered as immediate –under 15 minutes and below- and limited in time, the “epigenetic modifications” are durable and transmissible as traits to the following generations).

_the molecular mechanisms engaged are different and are not implying the same actors (Non-genomic effects are not related to a regulation of gene expression, all effectors are already present in the context without the need of a new transcription. Epigenetic consists in long term modifications of the chromatin defining the set of genes to be expressed; setting that is transmissible to the next generation).

For more details and appreciation of the “concept of non-genomic effect” please consider consulting the following reference:

Haller, J.; Mikics, E.; Makara, G.B. The Effects of Non-Genomic Glucocorticoid Mechanisms on Bodily Functions and the Central Neural System. A Critical Evaluation of Findings. Front Neuroendocrinol 2008, 29, 273–291, doi:10.1016/j.yfrne.2007.10.004.

We want to thank reviewer 2 for her/his comment that helped to correct our article

Reviewer 3 Report

The current manuscript submitted by Robin Mom et al, entitled as “Aquaporin-2 interaction with cortisol modulates its water permeability. Perspectives for corticosteroids non-genomic effects”. Authors investigated the effects of glucocorticoid cortisol on the water permeability of AQP2 and described the molecular mechanism of AQP2 inhibition during water transport by its direct interaction with the residues of the extra-cellular vestibule (in particular with R187 of the ar/R constriction), which lead to cortisol modifies the electrostatic potential of the conducting pore which in turn modifies the position of protein-water interaction sites and compromise the whole hydrogen bonds network. The manuscript is well written and experimental design is appropriate. The current version of manuscript accepted as it is for publication. I do not have any comments and enjoyed while reading it.

Author Response

We want to thank reviewer 3 for her/his appreciation of our work.

Reviewer 4 Report

This article is describing the investigation of Aquaporins (AQPs) transmembrane water channels for binding of corticosteroids via in silicomolecular dynamics simulations. According to the authors, naturally produced glucocorticoid cortisol inhibits channel water permeability.

Additionally, authors obtained the proof of new insight on the regulatory function of (AQPs) and the role of ions modulating the water permeability. The correlative analysis as illustrated on sequential 7 figures will constitute the important goals and potential novelty of this paper.

The following suggested changes and recommendations should be introduced before the publication of the manuscript.

1.     Page 2. Line 9 from the bottom is referring to Figure 1A, which is not included into the manuscript text. This should be corrected.

2.     Page 3, line 8, “cetone”?  Is this word referring to ketone group?

3.     Page 3 line 14 Figure 1 legend is included here but the original figure 1 is missing from the text. Please insert figure 1 in line 14.

4.     Page 18, line 8 Figure 7 legend is included here but the original figure 7 is missing from the text. Please insert figure 7 on page 18 on line 8.

5.      Page 21, line 2, “The in silico analysis provide some evidences of a realistic interaction”.  Is the term “realistic” an artificial as it should be defined as “specific” This should be clarified and edited!

6.     References, some of the references are missing doi. numbers.

The manuscript is of good quality and importance and is comprehensively written and edited in order to meet the standard for the articles to be published in International Journal of Molecular Sciences. Thus, I certainly recommend it for publication after the correction of these suggested minor changes. 

Author Response

The corrections suggested have been made. The text has been revised to remove the eventual typos. Track changes can be displayed in the word document.

We want to thank reviewer 4 for her/his comment that helped to correct our article.

Round 2

Reviewer 2 Report

The paper must be resubmited with lines number!!!!

Author Response

Please check the paper must with lines number.

Round 3

Reviewer 2 Report

Thank you for the new version.

I understand your point about non-genomic, but as for me, there are some confusions.

Maybe in future you can consider another name as

Rapid physiological signalling as decribed here:

https://pubmed.ncbi.nlm.nih.gov/18784332/

It will be nice to cited and decribe it in the manuscript.

Figure 2: please, change order: A shouls be up, and E must be down.

Figure 3, panel A: the colors is not very clear, at least in my screen.

Please, find other colors combination.

Panel D: I would suggest to use gray and black to see differences better.

Line 232: was performed and p.values were indicated.

Please, alos check punctuation: space between citaion and bracket etc.

Author Response

Thank you for your time and critical reviewing.

Indeed “rapid steroid signaling” could have been used as well to describe this type of physiological responses. We have added the suggested reference in the introduction and mentioned this denomination as synonymous with “non-genomic effects of corticosteroids”.

Figure 2 was rearranged to be readable in the portrait orientation as suggested.

The colors for figure 3 panel A have been changed. However, panel D colors were kept the same in order to maintain a coherent color code (“control” condition in gray and “COR” condition in purple) along the whole manuscript. 

We have parsed the text to remove the possible syntax troubles and you will notice that spaces were added before each citation bracket when needed.